# Mathematical modeling reveals ferritin as the strongest cellular driver of dietary iron transfer block in enterocytes

Joseph Masison[1]*, Pedro Mendes[1,2]*

**1** Center for Cell Analysis and Modeling, University of Connecticut School of Medicine, Farmington, Connecticut, United States of America, **2** Department of Cell Biology, University of Connecticut School of Medicine, Farmington, Connecticut, United States of America

* masison@uchc.edu (JM); pmendes@uchc.edu (PM)

## Abstract

Intestinal mucosal block is the transient reduction in iron absorption ability of intestinal epithelial cells (enterocytes) in response to previous iron exposures that occur at the cell scale. The block characteristics have been shown to depend both on iron exposure magnitude and temporality, and understanding block control will enable deeper understanding of how intestinal iron absorption contributes to pathological iron states. Three biochemical mechanisms implicated in driving the block behavior are divalent metal transporter 1 endocytosis, ferritin iron sequestration, and iron regulatory protein regulation of iron related protein expression. In this work, a model of enterocyte iron metabolism is built based on published experimental data that is capable of reproducing the mucosal block phenomena. The model is then used to estimate the quantitative contribution of each of the three mechanisms on the properties of the mucosal block. Analysis reveals that ferritin and iron regulatory proteins are the main intracellular mechanisms contributing to the mucosal block, findings congruent with experimental predictions. Lastly, DMT1 endocytosis is shown to play a role in limiting total iron uptake by enterocytes but does not contribute to the decrease in total iron transfer across their basal membrane seen in the mucosal block.

## Author summary

Dietary iron is absorbed by enterocytes, cells that line the walls of the initial part of the small intestine (duodenum). After being exposed to an iron dose, enterocytes show reduced capacity to absorb sequential iron doses for a certain length of time. Detailed understanding of this enterocyte iron absorption block helps better understand certain diseases, such as anemia and hemochromatosis. Three mechanisms have been implicated in this phenomenon: 1) endocytosis of divalent metal transporter 1 (DMT1), the protein that imports iron into enterocytes, 2) sequestration of iron in the cytoplasm by ferritin, and 3) regulation of ferritin expression by the iron regulatory proteins (IRPs), allowing enterocytes to produce more ferritin when iron levels are high. We build a computational model that reproduces these three mechanisms based on existing experimental data, and

**Data availability statement:** All relevant data and models are within the manuscript and its Supporting Information files. The model is also deposited in the BioModels Database as MODEL2405170001.https://www.ebi.ac.uk/biomodels/MODEL2405170001

**Funding:** We thank the National Institute of General Medical Sciences (NIH) for supporting PM's work on COPASI (grant R24 GM137787). The funders had no role in study design, data collection and analysis, decision to publish, or preparation of the manuscript.

use it to quantify the contribution of each mechanism to the block. Simulations show that ferritin and regulation by IRPs have the largest quantitative impact on the enterocyte iron block. DMT1 endocytosis does not significantly decrease the total iron transferred from intestine to the blood, but it limits the total enterocyte iron uptake, protecting the cell against high levels of free iron and the oxidative stress it would cause.

## Introduction

The duodenum, the first section of the small intestine, is the primary location for dietary iron absorption in mammals. Because iron loss is mostly passive, regulation of its entry into circulation through the enterocytes in the duodenum plays an important role in maintaining appropriate iron levels. The absorptive capacity of the duodenum needs to be able to increase or decrease according to the body's iron status. As great as a 20-fold increase in absorption has been observed [1] in response to anemia, compared to normal body iron levels. On the other hand, each enterocyte in the duodenum also needs to regulate the amount and state of its own internal iron, to protect against oxidative stress. Consequently, the regulation of iron absorption to match losses and prevent overload is crucial and has major implications for iron related diseases (e.g., anemia of chronic disease and hemochromatosis). Our knowledge of intestinal iron regulation continues to grow, which has led to the discovery and characterization of the process 'mucosal block' and the major regulatory role of the peptide hormone hepcidin.

The phenomenon that an initial oral iron dose leads to decreased duodenal absorption of subsequent doses, was observed experimentally as early as the 1940-1950s [1,2]. Since then, there have been many efforts to understand the regulation driving mucosal iron absorption [3,4]. Iron stores in the mucosa were hypothesized to impact iron absorption, with absorption increasing only if those stores dropped. After the hormone hepcidin was discovered [5–8], it was shown that many aspects of mucosal iron absorption behavior could be explained by hepcidin's action (part of a systemic negative feedback loop, also modulated by erythropoietic [9–11] and inflammatory signals [12], but ultimately driven by global iron levels). Subsequently, there has been a distinction between the systemic mucosal iron absorption block mediated by hepcidin and a cellular intestinal block that is dependent on the biochemistry of individual enterocytes [3,13–15]. This enterocyte-level block is revealed by *in vitro* systems lacking hepcidin and by the inability of hepcidin to explain all intestinal iron absorption behavior [16,17]. In this work, we use computational modeling to estimate the quantitative impact of distinct cellular mechanisms within individual enterocytes to understand the mucosal block, specifically as it relates to how an iron dose alters subsequent enterocyte iron absorption. While our focus is on the cellular mechanisms, still the systemic effect of hepcidin has to be considered as enterocytes *in vivo* are always affected by it.

The mucosal block has been suggested to depend on three distinct biochemical mechanisms. The first of these is endocytosis of the divalent metal transporter 1 (DMT1). DMT1 is the main importer of inorganic ferrous ($Fe^{2+}$) iron located in the enterocyte apical membrane. After exposure to iron, fractions of DMT1 on the luminal membrane are endocytosed [18,19], which temporarily decreases the number of importers and thus potentially decreases the capacity to absorb iron, while keeping total DMT1 protein amount (the plasma membrane fraction plus the endocytosed cytoplasmic fraction) relatively stable [19]. The second mechanism proposed is the post transcriptional regulation of iron related proteins through iron regulatory proteins 1 and 2 (IRPs) [20,21]. IRPs bind sequences in the untranslated

regions of mRNA called iron-responsive elements (IREs) to either increase (when in the 3' end) or repress (in the 5' end) translation. In low iron conditions IRPs cause increased DMT1 synthesis and decreased synthesis of ferroportin (FPN), the basolateral exporter of iron, and of ferritin (FT), a protein used to store iron intracellularly, among others. Under iron excess, IRPs unbind from the target mRNAs leading to opposite effects on the levels of those proteins. FT synthesis is controlled by IRPs, but there is also iron depleted FT in enterocytes at baseline. As iron enters the cell, the preexisting FT is able to buffer this iron independently of regulation by IRPs. No quantitative descriptions exist about the relative importance of immediate FT buffering versus FT concentration changes directed by IRPs in response to repeated iron doses. Therefore, a third possible mechanism is the buffering effect that existing FT has on cytoplasmic iron levels. In summary, three mechanisms appear to co-exist that could lead to the observed hepcidin-independent mucosal block: 1) endocytosis of DMT1, 2) regulation of protein levels by IRPs, and 3) the buffering effect of FT, but the relative importance of each one is not known. Here, a computational model is constructed that incorporates all three mechanisms for the specific purpose of quantifying the contribution of each of these mechanisms to the cellular mucosal block.

## Results

### A model of enterocyte iron absorption block

The kinetic model developed here is composed of a set of ordinary differential equations encapsulating the processes of iron uptake from the intestinal lumen, intracellular iron metabolism, and export across the basolateral membrane for an individual duodenal enterocyte. It is built for the specific purpose of estimating the quantitative impact of three mechanisms (DMT1 endocytosis, FT iron sequestration, and IRPs regulation of FT expression) on the dynamics of enterocyte iron absorption and their contribution to the cellular mucosal block. The model includes five cellular compartments: lumen, apical membrane, intracellular space, basolateral membrane and plasma. The model tracks the following iron species: ferrous luminal iron, "free" ferrous cytosolic iron (labile iron pool, LIP), ferric and mineralized iron inside FT, iron exported into plasma, and iron sequestered by the rest of the body. In addition, it contains proteins that transform or bind these iron species. The model has a total of 13 distinct chemical species and 19 reactions (Fig 1). These reactions capture the three mechanisms studied as well as FPN-dependent iron export and its regulation by hepcidin. FPN regulation is widely accepted as the main mechanism for the systemic regulation of iron absorption. FPN is regulated through hepcidin which, in non-pathogenic conditions, is reflective of the whole-body iron level. The first three mechanisms may contribute to the enterocytic iron block and are the main subject of this research. The ferritin iron sequestration component was the subject of a previous publication that provided it [22] as a "model brick" [23] (MODEL2211030001 in BioModels Database), which is appropriately re-used here.

All models, by necessity, carry assumptions and simplifications and that is true in the present one. Specifically, 1) when protein synthesis is included this is represented by only one reaction producing the mature protein, 2) heme iron uptake is not considered in this model, 3) the role of the ferroxidases DCYTB and HEPH are not included explicitly, 4) only ferritin is post-translationally regulated by IRPs, 5) the distinct action of IRP1 and IRP2 are aggregated into a single variable "IRPs". Further details about these assumptions can be found in Methods.

The model contains a total of 68 parameter values, including compartment sizes (5), species initial concentrations (13), reaction rate parameters (43) and other global parameters (7). There are 19 additional parameters introduced to simulate experimental

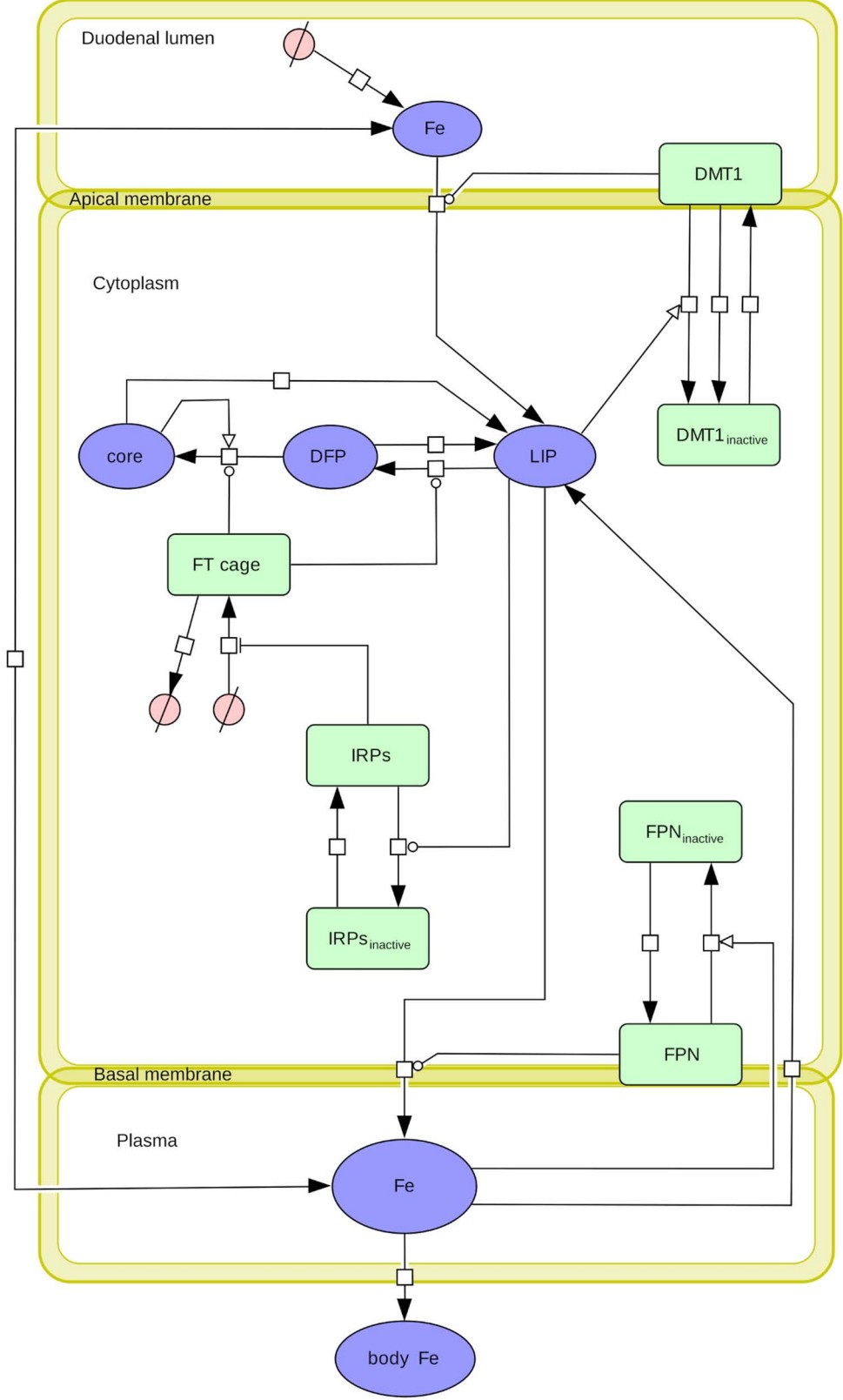

**Fig 1. Enterocyte iron absorption model, SBGN process diagram.** A model encapsulating iron transport from the intestinal lumen through a single enterocyte was constructed using COPASI [24–26]. Species inclusion

is motivated by the intent to capture a minimal, but sufficient detail needed to reproduce the mucosal block phenomenon in duodenal enterocytes. The model has 5 compartments (lumen, cytoplasm, and blood, which are three-dimensional, and the apical and basal membranes, which are two-dimensional). Thirteen model variables are included to track the concentrations and densities of the relevant chemical species. Abbreviations: DMT1: divalent metal transporter 1, FT: ferritin, DFP: diferric peroxo complex ferric intermediate (ferric iron in FT), core: FT mineralized core iron, IRPs: iron regulatory proteins, FPN: ferroportin, LIP: labile iron pool, Fe: iron (lumen, blood, and body forms). Figure created with Cell Designer [99] and Inkscape, adopting the Systems Biology Graphical Notation standard (SBGN) [100,101].

procedures, including scheduling, execution, and tracking of lumen iron addition events. For model calibration, 55 model parameters were determined directly from published experimental data, and their provenance, rationale, and calculations are documented in the Methods section. The 13 parameter values that were not available in the literature were estimated to match published experimental results [4,13,27–29]. For example, since several of the DMT1 endocytosis regulation parameters were not measured directly, experimental time courses of DMT1 cellular location in response to iron exposure [27,28] were used to determine those parameter values. Parameter estimation was carried out by nonlinear least-squares on the numerical solutions of the full ODE model, using the parameter estimation task in the software COPASI [24–26] (using a combination of optimization algorithms: Genetic algorithm, Hooke and Jeeves, and Particle swarm). The result was a model that fits the experimental observations (Fig 2A-2C). Additionally, the steady state apical and cytosolic DMT1 concentrations exist in the fitted model in a ratio of 3.12:1.88, which is similar to the 3:2 shown in Tandy et al. 2000 [29]. For the simulation of experiments some of the parameters were then scaled appropriately (e.g., the lumen compartment size to account for experiments done with a population of cultured cells versus an individual cell). The models used for parameter estimation and their experimental data can be found in the S1 Data. The full set of equations used in the model are listed in S1 Text, and the numerical parameter values in S2 Text.

## The model exhibits mucosal block behavior

Model *validation* tests if the model produces appropriate results compared to data gathered in conditions outside those used for fitting of the model. Data from three studies that measured enterocyte iron kinetics [14,15,30] were set aside *a priori* and not used at all in model calibration. The first two used cultured Caco-2 cells, a widely-used *in vitro* system for studying enterocyte iron absorption [31–33], and tested dose dependent iron absorption over time. The third used live Sprague-Dawley rats and *in vivo* duodenal isolation and dosing to look at the temporal aspect of mucosal block. Without modifying any parameters, other than the initial conditions reflective of each experiment, the model adequately reproduced the observations of the validation experiments (Fig 2D-2F). The fact that the datasets tested conditions outside the calibration range (lower iron dose for the second dataset [30] and much longer timescales for the third [15]) adds confidence to the model performance. To clarify the intricacy of the methods of Frazer et al. 2003 [15], (Fig 2E and 2F) each time point represents the cumulative effect of two iron doses and the time between them. For each, a rat was given an intragastric dose of iron and then, after the designated time, a second test dose. After 30 minutes, the percentages of the second dose that were 1) absorbed by the duodenum (uptake) and 2) exported out into the rest of the body (transfer) were measured compared to the total dose. In this way the authors measured the percentage of the second dose that was absorbed (i.e., the "mucosal block") as a result of the first dose and the time between them. The results are plotted in Fig 2E and 2F.

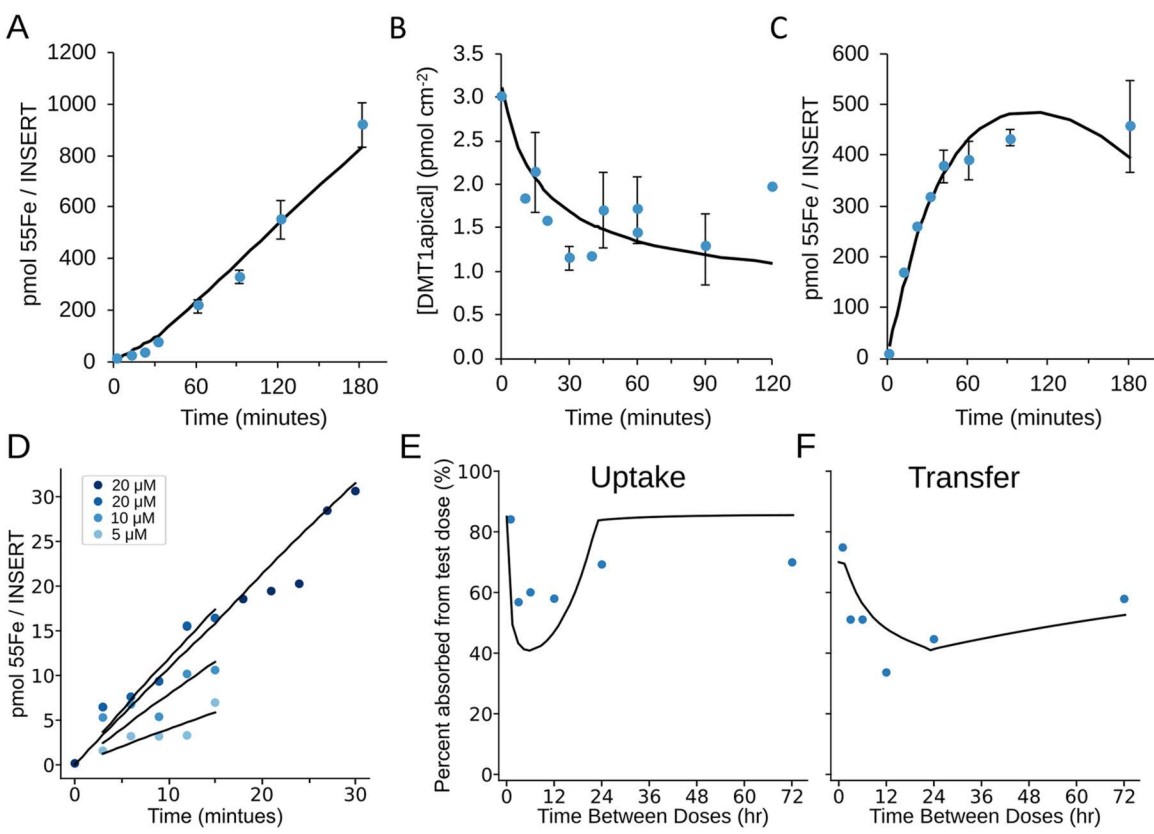

**Fig 2. Model calibration and validation.** The model was parameterized using kinetic constants from published experiments. For those with no literature value (DMT1 endocytosis free/LIP modified, DMT1 fusion, FPN and IRPs activation/inactivation reaction parameters listed in S2 Text), were estimated by nonlinear least-squares regression over the calibration data (see Methods) (A-C). Validation then checks whether the model with the estimated values behaves in agreement to a number of independent experiments (D-F). In all plots the model output is represented by a black line, and the experimental data by blue circles. Error bars are from the publications that presented the experimental datasets. (A) Iron exported through FPN over time following an apical iron dose (20μM) [13]. (B) Time course of DMT1 endocytosis triggered by an apical iron dose (20μM) from two independent experiments [27,28]. (C) Cellular iron over time following an apical iron dose (20μM) [13] (A-C calculations carried out with file parameter_estimation.cps in S1 Data). (D) Dose-dependent cellular iron uptake over time, from Fig 2 of Colins et al. 2017 [30] and Fig 6A of Cegarra et al. 2019 [14] (file validation_cegerra.cps in S1 Data). (E) Percent uptake and (F) percent transfer of duodenal iron dependent on time (horizontal-axis) between iron doses from Fig 1 of Frazer at al. 2003 [15], (file validation_frazer.cps in S1 Data). "Transfer" represents iron absorbed into the body, *i.e.,* first taken up by enterocytes and then exported to plasma.

The model simulation generally matches the experimental observations, especially for the iron time courses (Fig 2D). For the whole animal however, while the model captures the same block behavior overall (qualitatively similar curves, similar minimum absorption and block alleviation times – around 10 and 24 hours respectively), there are some differences; the model displays a higher absorbance recovery than the experiment (at 72 hours it predicts a greater absorption percentage, Fig 2E) and also a deeper uptake (greater uptake reduction) and shallower transfer (Fig 2E and 2F). When evaluating the model performance, it is important to note the model is built with the explicit purpose of being minimal, but sufficient in capturing the block at the *individual enterocyte* scale. As a result, because the experiment is done in a whole animal system, there are several other processes contributing to iron absorption behavior, especially considering the multiple day time scale. The intestinal microbiome, cell viability/death as a result from the stress and length of the experiment (which involved surgical isolation of the duodenum), and presence of non-enterocyte cells within the duodenum

are just a few processes that would affect iron absorption measurement. The model, however, assumes 100% cell viability and is composed only of enterocytes. Thus, these slight deviations of the model from the data appear to be acceptable.

## Mediators of mucosal block

Having obtained a model that adequately fits experimental data, even in conditions at a vastly different time scale and iron dose from those it was built from, the model was used to explore the quantitative impact of each of its components on mucosal block. The impact analysis is supplemented by time course simulations providing possible explanatory mechanisms underlying the results.

**Steady states.** While enterocytes are never in a steady state, given the perturbations caused by meals, it is useful to establish a steady state that would exist if the average iron from meals was continuously infused. Such steady states provide the average values of each variable and will be used as initial conditions for time courses. First, a reference steady state of the model was determined (Table 1) by setting a constant concentration of iron in the lumen and running the model for $5.5 \times 10^6$ seconds from the initial state (over 60 days). This steady state was then used as the basal state that will become the initial condition for subsequent simulations. This simulated steady state display concentrations that are within biologically feasible ranges. A deficient iron diet steady state was determined by fixing "$Fe_{lumen}$" at a value of 0. In this iron deficient condition, all of the iron species in the model approach 0 except for total body iron (where the iron is exported to). Finally, an iron-overload diet was simulated by fixing the lumen iron at 125 nM. Iron overload results in increased FT and FT-associated iron and a reduction of membrane FPN (a result of hepcidin regulation). Under both dietary conditions, the model behaves as expected regarding the direction change in iron species concentrations; FT, IRPs, and FPN all match experimentally predicted changes as well.

Next, the model is used to quantify the impact of DMT1 endocytosis, FT iron sequestration, and IRPs regulation on the dynamics of enterocyte iron mucosal block. There are two categories of interest in the mucosal block: iron *transferred* (the total amount of iron passed through the enterocyte to the rest of the body, consisting of: $Fe_{Blood} + Fe_{body}$) and iron *uptake* (the total iron imported by the enterocyte from the lumen, consisting of $Fe_{Blood} + Fe_{body} + LIP + DFP + FT$ core iron). When describing the block, a "larger" block magnitude means

**Table 1. Simulated steady-state concentrations at different iron loadings.** Steady-state concentrations at each fixed luminal iron level: basal at 12.5 nM, low iron at 0 M (Low Fe diet), and high iron at 125 nM (High Fe diet). APC ("atoms per core") is the average number of iron atoms trapped in each ferritin cage.

| Species | Location | basal | Low Fe diet | High Fe diet |
|---|---|---|---|---|
| Fe | Lumen | 12.5 nM | 0 M | 125 nM |
| FT cage | Cytoplasm | 2.38 nM | 2.18 nM | 10.8 nM |
| core | Cytoplasm | 0.368 μM | 0 M | 46.6 μM |
| APC | Cytoplasm | 1550 | 0 | 4299 |
| DFP | Cytoplasm | 0.13 nM | 0 M | 590 nM |
| LIP | Cytoplasm | 0.12 μM | 0 M | 20.1 μM |
| DMT1 | Apical membrane | 3.12 pmol cm$^{-2}$ | 3.12 pmol cm$^{-2}$ | 2.94 pmol cm$^{-2}$ |
| DMT1 | Cytoplasm | 1.88 pM | 1.88 pM | 2.06 pM |
| Fe | Blood | 5 nM | 0 M | 46.7 nM |
| FPN | Basal membrane | 0.1 pmol cm$^{-2}$ | 0.1 pmol cm$^{-2}$ | 0.053 pmol cm$^{-2}$ |
| FPN | Cytoplasm | 0 M | 0 M | 0.047 pM |
| IRPs | Cytoplasm | 68.9 pM | 76.2 pM | 4.15 pM |
| IRPs inactive | Cytoplasm | 7.3 pM | 0 M | 72 pM |

that there is *less* iron uptake/transfer, so the maximum block will have the minimum iron uptake/transfer values. Thus, to quantify the block, five properties of the block were chosen as a base for analysis (shown in the legend for Fig 3): 1) the maximum block magnitude for *uptake* (Minimum Uptake Value), 2) the time at which the maximum block occurs for *uptake* (Minimum Uptake Time), 3) the time at which the block in *uptake* is recovered to >95% of its initial value (Recovery of Uptake Time), 4) the time at which the maximum block occurs for *transfer* (Minimum Transfer Time), 5) and the maximum block magnitude for *transfer* (Minimum Transfer Value). These five properties were monitored in various simulations that were designed to analyze the effects of various perturbations, namely the size of the iron dose, the inhibition of FPN iron export by hepcidin, rate of DMT1 endocytosis, levels of iron sequestration by FT, and strength of the regulation of FT expression by IRPs.

**Dose size modifies all block characteristics.** Mucosal block is dependent on the magnitude of the initial dose of iron [15]. To test the effect of the loading dose on block severity, simulations were run varying the initial loading dose in the range 1.5 - 5 μM while keeping the test dose unchanged (at 2.5 μM) (Fig 3). Simulations show an increased loading dose increasing all five block characteristics, thus causing a more severe block. (Fig 3E and 3G). The magnitude of the simulated difference in block between the lowest and highest dose – around 3.5-fold for uptake (Fig 3A), and 2-fold for transfer (Fig 3C) – as well as the linear relationship between dose size and block depth (Fig 3E) agree with experimental data (also linear with 4-fold block increase from loading dose variation, see Fig 6 of Frazer et al. 2003 [15]).

**Hepcidin control of FPN determines iron transfer.** The next mechanism tested, also with an effect well characterized in literature, is the hepcidin-FPN axis. As mentioned in the Introduction, hepcidin levels increase in response to rising blood iron levels and subsequently reduce the export potential of enterocytes by decreasing their FPN basal fraction. More hepcidin driven degradation of FPN leads to greater FPN activity reduction so less transfer for the same iron dose should be observed. The rate constant ($k_{cat}$) for the FPN inactivation reaction in the model represents the magnitude of the hepcidin response relative to the same change in blood iron levels. When the $k_{cat}$ for the FPN inactivation reaction is varied (0.7 - $2.8 \times 10^{-6}$ s$^{-1}$), keeping all other parameters fixed, the simulations show an increase of the characteristics of the transfer curves (Fig 3D). This trend reveals that in the model, the more sensitive the degradation of FPN is to the level of hepcidin (and thus the level of plasma iron), the smaller the amount of iron transferred to the plasma is. Interestingly, when the uptake curve is analyzed (Fig 3B), there are mostly minimal changes, with the largest effect on the recovery time, though still relatively minor (Fig 3F and 3H). This result indicates that FPN regulation by hepcidin controls the enterocyte iron export, but the enterocyte still maintains the same total iron uptake capacity in the model, regardless of its ability to export iron to the plasma.

These loading dose and FPN related results confirm that the model behaves according to the known physiology. The next steps are to use this model to study the contribution of the intracellular enterocyte mechanisms, which have not been estimated otherwise, and are the main objective of this study.

**Sensitivity of DMT1 endocytosis to the LIP determines iron uptake but not transfer.** Simulations were carried out where the rate constant ($k_{cat}$) of the LIP-induced DMT1 endocytosis was varied in the range 1-40 s$^{-1}$, to assess the impact on the mucosal block (Fig 4). This rate constant represents the sensitivity of endocytosis to LIP, in other words how much DMT1 is sequestered into the cytoplasm from the apical membrane relative to the same LIP change. A higher value of this rate constant causes more endocytosis for the same amount of LIP change. A higher value of this rate constant leads to a higher maximum reduction in

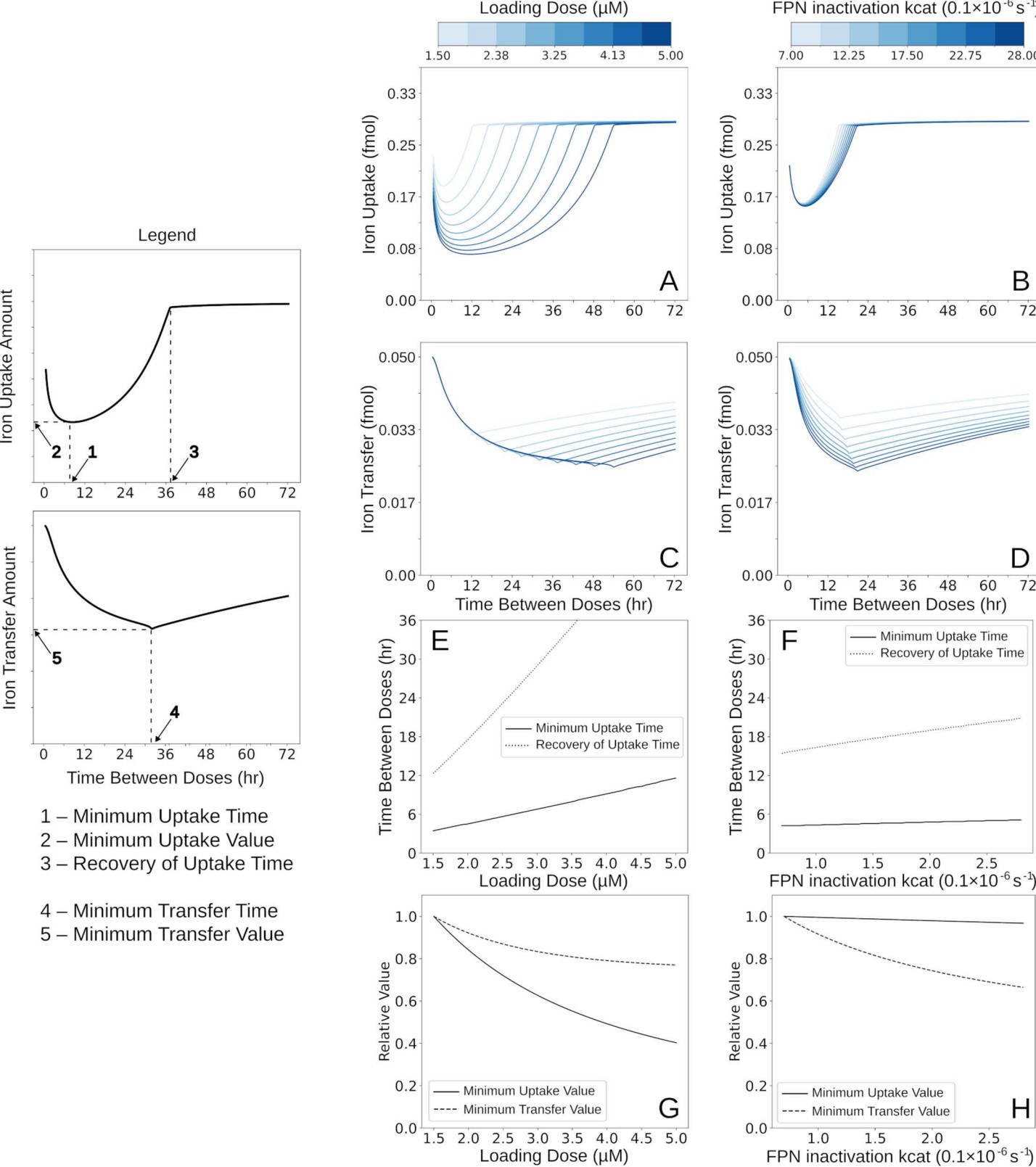

**Fig 3. Dose size and hepcidin control of FPN reproduce known effects on mucosal block characteristics.** The effect of loading dose and sensitivity of FPN basal fraction to blood iron levels on mucosal block were tested. Simulations were run for nine iron doses (1.5 - 5 µM) and nine values of FPN inactivation reaction rate constant

($k_{cat}$, 0.7 - 2.8×10⁻⁶ s⁻¹, baseline, 1.4×10⁻⁶ s⁻¹). The abscissa represents the time interval between loading and the test dose, while the ordinate represents the amount of iron absorbed 30 min after application of the test dose (2 µM). For each dose variation, 300 simulations were run with different time intervals (0-72 hrs). (A, B) Uptake. Uptake iron amount (fmol) based on dose (A) and $k_{cat}$ (B) and time between dose is plotted. (C,D) Transfer. Transfer iron amount (fmol) based on dose (C) and $k_{cat}$ (D) and time between dose is plotted. (E-H) Block characteristics. In each panel, a subset of the five block characteristics are plotted as a function of the loading dose value (E,G) or the $k_{cat}$ value (F,H).

iron uptake (Fig 4A), while the other block characteristics stay essentially unchanged (Fig 4E, 4I, and 4M). This means that variations in the rate of DMT endocytosis significantly alter iron uptake by the enterocyte, but do not affect how much iron is subsequently transferred to the plasma, after being taken up by the enterocyte from the lumen. In other words, the block severity seen in iron transfer to plasma is not explained by DMT1 endocytosis.

**FT synthesis impacts both uptake and transfer blocks.** Finally, we tested the impact of post-transcriptional regulation of FT expression by IRPs on mucosal block (Fig 4). These examine the interpretation of experiments by Galy *et al.* that IRPs and FT are responsible for the mucosal block [34]. Low intracellular iron (LIP) leads to increased activity of IRPs, which then repress the rate of FT synthesis, eventually leading to a state of low FT concentration. Conversely, when LIP raises, the activity of IRPs is reduced, leading to higher FT concentration. Reflective of this biology, in the model the rate of FT synthesis is expressed as a fixed basal rate plus a component that depends on the concentration of active IRPs. As the value of the rate constant increases, the relative effect on FT synthesis due to the IRPs increases. In other words, if the basal rate value is high, then the same decrease in IRPs activity will have a greater effect on the magnitude of the increase in FT concentration. The effects on uptake and transfer when $k_{cat}$ values were varied (0.035-1.5 pMs⁻¹) supports the interpretation of Galy *et al.* 2013 [34]. As the $k_{cat}$ increases, the uptake block is alleviated (Fig 4B), while the transfer block increases in severity (Fig 4F).

The active IRPs modulation of FT expression is also dependent on the value of the FT expression $K_m$, which sets the range for which the rate of FT synthesis is sensitive to changes in IRPs. What simulations showed is that the effect of varying this $K_m$ value is dependent on the FT expression basal rate, $k_{cat}$. To best characterize this effect, multiple sets of simulations with different values of basal expression rates were run, in each one, the $K_m$ is set at different values between 7 pM and 28 pM (14 pM is the base value); starting with one with the minimum basal rate of 0.035 pMs⁻¹ up to one with the max of 1.5 pMs⁻¹. What is seen as the $K_m$ increases is that the uptake and transfer block deepens (larger block) in a time-dependent way, with larger effect at larger time intervals between doses (with similar block characteristics at early time intervals, 0-12 h, regardless of $K_m$ value). Importantly, the effect of the $K_m$ variation is secondary to the basal rate. Fig 4D, 4H, 4L and 4P depicts the results of varying the $K_m$ between 7 pM and 28 pM with $k_{cat}$ of 1.5 pMs⁻¹ (the maximum basal rate value simulated), due to the observed maximum effect of $K_m$ variation at that expression rate value. Lower expression values show similar block characteristics effects, but with progressively smaller magnitude, with almost no variation in block characteristics with $K_m$ variation at basal expression rate of 0.035 pMs⁻¹. This result is due to the fact that the IRPs indirectly modulate FT concentration through the FT expression, so the degree to which they impact FT levels depends on the magnitude of the rate that they are modifying. In other words, the same relative IRPs dependent reduction in FT expression rate will result in a greater absolute change in FT protein level when the absolute expression rate is higher.

**FT concentration affects both uptake and transfer blocks.** Because the regulation of IRPs affects the FT levels, we independently tested the effect of FT concentration, independent of IRPs regulation. In this case we fix the total concentration of FT in the model and set it at different values. The motivation behind this is that FT concentration in

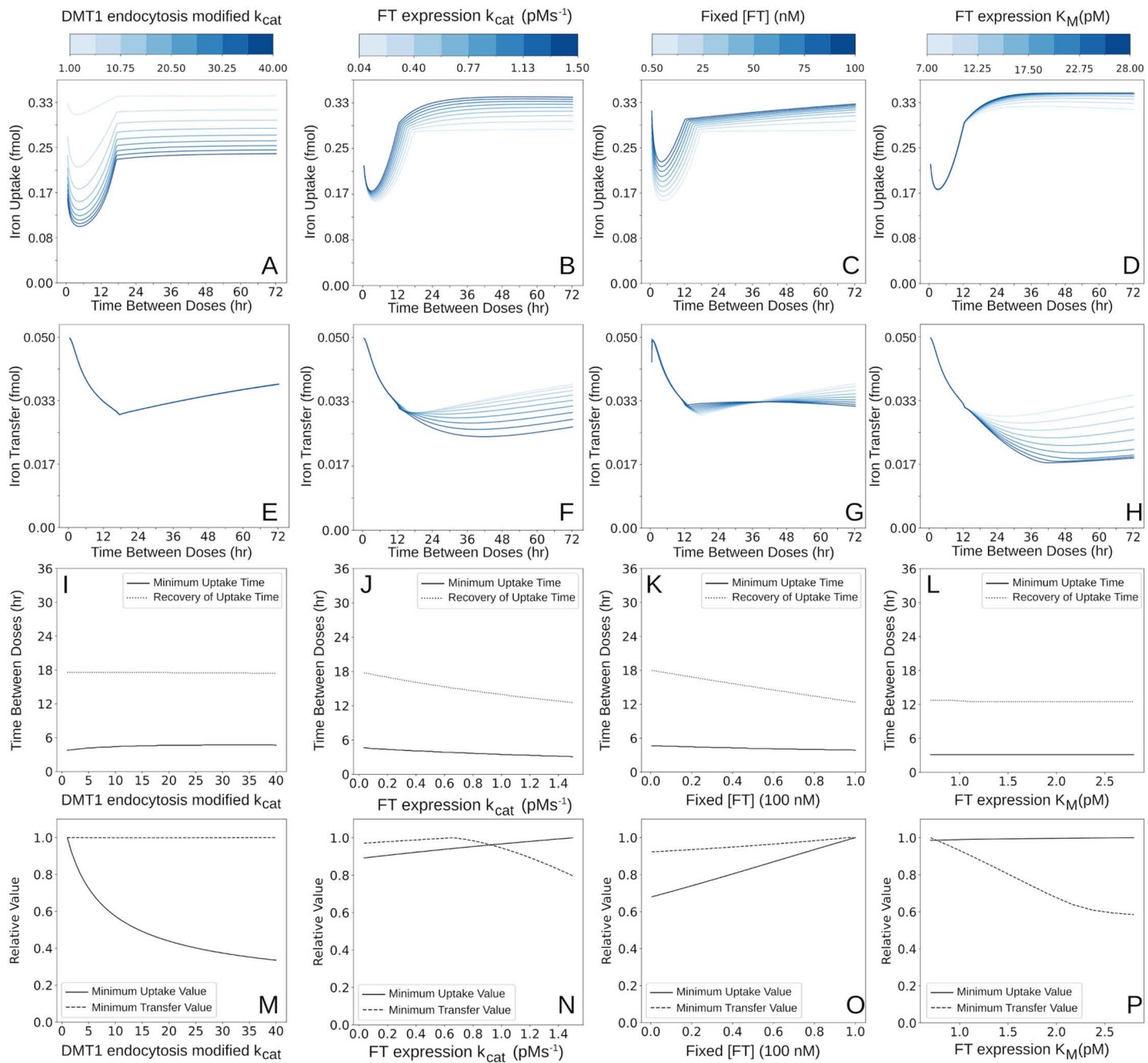

**Fig 4. Comparison of the effects of FT, IRPs regulation, and DMT1 endocytosis on mucosal block.** The reaction rate constant ($k_{cat}$) for the DMT1 endocytosis, which is activated by LIP, was varied between 1 s$^{-1}$ and 40 s$^{-1}$ (baseline 14.5 s$^{-1}$); the effect of LIP on DMT1 endocytosis is higher at higher values of this parameter. The rate constant ($k_{cat}$) for the synthesis of FT, which is regulated by IRPs, was varied in nine steps between 0.035 and 1.5 pMs$^{-1}$ (baseline 0.0768 pMs$^{-1}$); the effect of IRPs on FT synthesis is higher at higher values of this parameter. Next, we conducted simulations where the FT concentration is fixed at values in between 0.5 nM and 100 nM (baseline, 2.375 nM); this probes the effect of FT iron buffering independent of the feedback loop through the IRPs regulation. Lastly, we conducted simulations where the FT expression $K_m$ is set at values between 7 pM and 28 pM (baseline, 14 pM); this probes the sensitivity of FT expression to IRPs level changes. In all panels the abscissa represents the time interval between the two iron doses. Panels A, B, C and D show the effect of the four parameters on the total iron uptake (expressed as fmol of iron). Panels E, F, G and H show the effect of the parameters on the total iron transferred (also expressed in fmol of iron). Panels I, J, K, and L show the time interval at which the block is most effective as a function of the parameter value (the Minimum Uptake Time of panels A, B, C, and D), as well as Recovery of Uptake Time. Panels M, N, O, and P express values of the Minimum Uptake Value, and Minimum transfer value as a function of parameter value.

a terminally differentiated enterocyte is a consequence of the history of iron status during its differentiation. Increased mature enterocyte FT level has been shown to result in part from increased iron exposure during differentiation [35]. Thus, fixing the concentration of FT at different values simulates different histories of iron exposure during differentiation. Simulations (Fig 4) show that a higher concentration of FT decreases the maximum uptake reduction (less block), while the time interval at which there is maximum uptake reduction does not change. A higher concentration of FT also reduces the time needed to recover iron uptake (Fig 4C). For transfer, the time of maximum block decreases with increased FT concentration (Fig 4G and 4O). Around the 36-hour dosing time interval there is an inversion in the effect on the trend. An increase in FT concentration leads to less of a block up to those time intervals, but after, it leads to more (Fig 4G and 4O), though this effect is of a small magnitude.

**Sensitivity analysis.** Fig 4 is useful to assess the effects of each parameter variation on the mucosal block, with additional support by a formal sensitivity analysis for those parameters. Sensitivity analysis quantifies the magnitude of the effect of a parameter change on the value of a model variable of interest. These can be expressed as partial derivatives of the effect of the parameter on the variable:

$$s(V, p) = \frac{dV}{dp}, \tag{1}$$

where $V$ is the variable and $p$ the parameter. A more useful expression can also be used in terms of relative changes:

$$R(V, p) = \frac{dV / V}{dp / p} = \frac{dlog(V)}{dlog(p)}. \tag{2}$$

The values of $R$ can easily be compared across variables and parameters, while those of $s$ depend on the magnitudes of $V$ and $p$ [36]. Here we estimate the values of $R$ where the variable of interest are the various block characteristics and the parameters are those already studies in Fig 4. The analysis is carried out by first running a simulation with the initial conditions reflecting the basal state of Table 1 and the baseline values of each of the five characteristics of the block were determined. Then additional simulations were run where each parameter was increased by 10%, while keeping the other parameters at baseline value, and the resulting relative change in the model output for each of the five characteristics was determined (Table 2). Because relative changes are measured, all values can be compared directly with each other. For example a 10% increase in dose lengthens recovery time by 12.5% and lowers uptake value by 6.2% (increases uptake block). The sign of these sensitivities indicates the direction of change in the block characteristic after a 10% increase in that parameter (negative meaning that the variable reacts in the opposite direction of the change in the parameter). In addition to the five parameters varied above, several other model parameters were included in this analysis.

The results of the sensitivity analysis reaffirm the trends of Figs 3 and 4, additionally, it reveals that the uptake mucosal block characteristics are most sensitive to the magnitude of the loading dose. Next is the effect of FPN inactivation, and FPN parameters in general, which dictate the most sensitive impact on transfer characteristics. The sensitivity to FPN is reflective of the known importance of hepcidin in regulating intestinal iron absorption. Of the three other mechanisms tested the largest effect on the transfer block are parameters affecting FT level ("Fixed [FT]" and IRPs expression regulated via "FT expression $k_{cat}$" and most potently by "FT expression $K_m$"). To reemphasize the importance of FT value and regulation by IRPs,

**Table 2. Sensitivity of mucosal block to model parameter variations.** The relative change in each of the five mucosal block characteristics resulting from a simulation run at baseline compared to a simulation where a single parameter value is increased by 10% of its baseline value are shown ("-" indicates negligible change). Compared are: 1) the time at which the maximum block occurs for *uptake* (Minimum Uptake Time), 2) the value of the maximum reduction in *uptake* (Minimum Uptake Value), 3) the time at which the block in *uptake* is lifted (Recovery of Uptake Time), 4) the time at which the maximum block occurs for *transfer* (Minimum Transfer Time), 5) the value of the maximum reduction in *transfer* (Minimum Transfer Value).

|  | Minimum Uptake Time | Minimum Uptake Value | Recovery of Uptake Time | Minimum Transfer Time | Minimum Transfer Value |
|---|---|---|---|---|---|
| Loading dose | 0.95 | -0.62 | 1.25 | 1.24 | -0.25 |
| FPN inactivation $k_{cat}$ | 0.16 | -0.02 | 0.20 | 0.22 | -0.29 |
| FPN inactivation $K_m$ | -0.32 | 0.05 | -0.51 | -0.49 | 0.79 |
| FPN activation $k_{cat}$ | – | – | -0.02 | -0.02 | 0.05 |
| FT expression $k_{cat}$ | – | 0.01 | -0.02 | -0.02 | – |
| FT expression $K_m$ * | – | 0.01 | -0.03 | 0.66 | -0.44 |
| IRPs degradation | – | – | – | – | – |
| FT turnover | – | 0.01 | -0.02 | – | – |
| Fixed [FT] | 0.16 | -0.05 | 0.16 | 0.16 | -0.03 |
| DMT1 $k_{cat}$ | 0.08 | -0.37 | – | – | – |

*FT expression $k_{cat}$ of 1.5 pMs$^{-1}$.

with the similar transfer sensitivity as FPN related parameters. DMT1 endocytosis has an effect on the iron uptake, but little effect on the iron transfer.

**Time course analysis of iron uptake and transfer.** To provide insight into how DMT1 endocytosis and FT synthesis impact the block, time course simulations were run for a duration of 72 hours. Rather than assessing the block at different dose intervals, here we use one fixed dose interval of 12 hours, using a dose of 2.5 µM iron in the lumen, and then follow the dynamics of iron distribution across all model compartments. We examine the effect of changing the same parameters (as above, Fig 5). A control baseline time course (Fig 5A) shows the iron added to the lumen enters the cell over the course of 12 hours and mainly partitions to LIP, raising the LIP by orders of magnitude. Throughout, LIP is exported through FPN into the blood and consequently sequestered into the body. By 48 hours, there are only trace amounts of LIP export. Fig 5C-5E shows the effect on FT core iron, LIP, and transferred iron when simulations are run under the same conditions as the control, but now varying the parameters for rate of FT synthesis or the (LIP-induced) DMT1 endocytosis. Increasing the rate of FT synthesis decreases the total iron transferred, especially in the last 36-72 hrs (Fig 5C and 5E), in concordance with the block analysis (Fig 4E and 4K). The additional information revealed by the time course is that the majority of iron not transferred is sequestered as FT core iron (Fig 5C), and that increased FT concentration also increases LIP buffering (Fig 5E). This means the more active the FT synthesis is (e.g., by IRPs suppression and disinhibition), the more the enterocyte can respond with new FT synthesis, and the mucosal block is more pronounced. For DMT1, however, increasing its endocytosis sensitivity to LIP results in a dramatic reduction of the LIP peak (Fig 5F), there is little effect on the fraction of iron in the FT core or the amount of transferred iron (Fig 5D).

**Direct comparison of DMT1 and FT/IRPs role in cumulative mucosal block (terminal time point analysis).** To better visualize the effects of varying the rate of (LIP-induced) DMT1 endocytosis and the rate of (IRPs inhibited) FT synthesis, the simulated terminal time point values of uptake and transfer (representing cumulative iron uptake/transfer) were plotted against their rate constant values (Fig 5B). This more clearly shows how, at 72 hours, increased FT expression allows more iron uptake (+3%, potentially due to the increased LIP buffering ability shown in Fig 5E) while simultaneously decreasing the total iron transfer

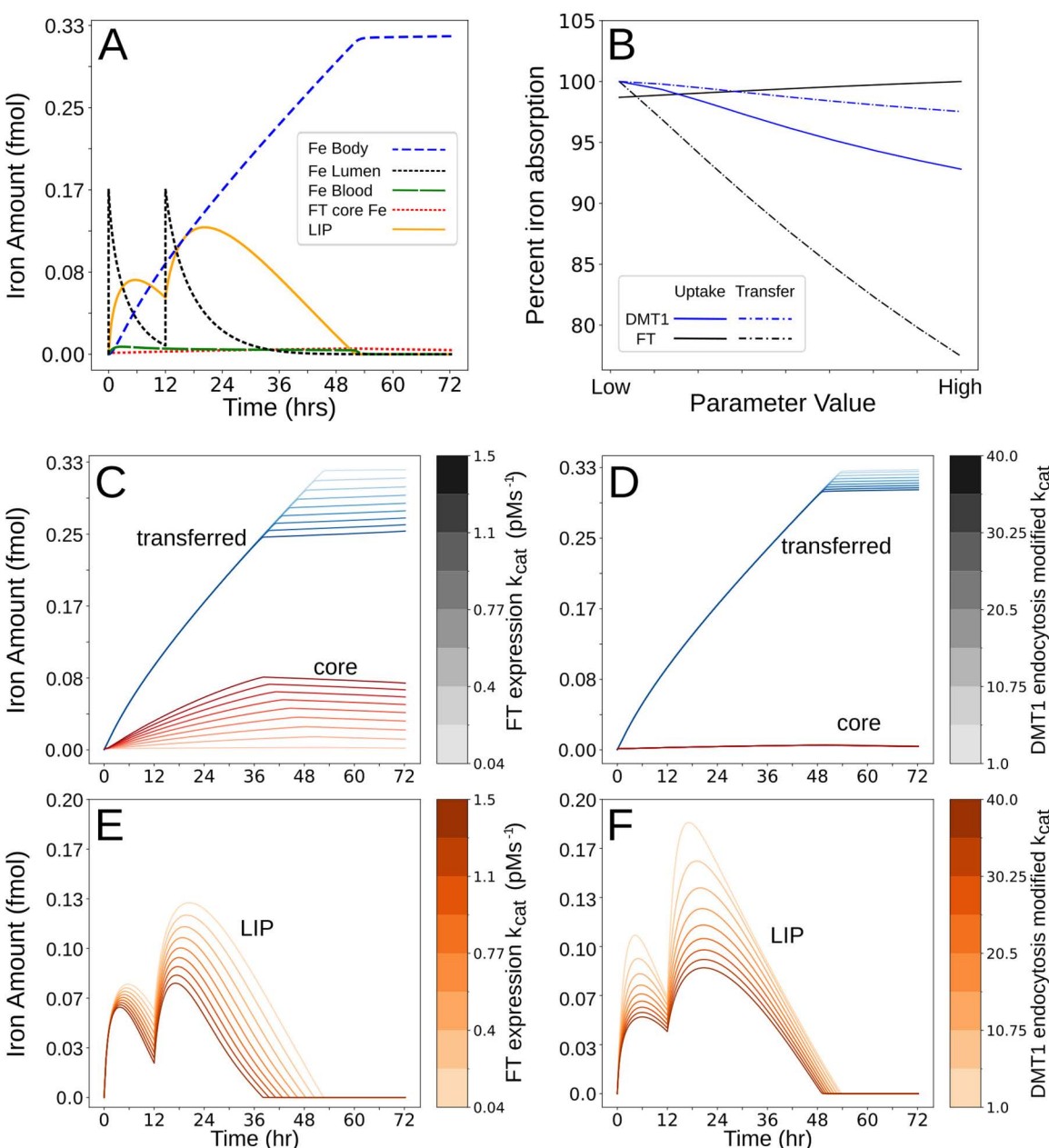

**Fig 5. Time course analysis of iron uptake and transfer.** (A). A baseline (control) simulation was run to show temporal iron species dynamics in response to a loading dose (time 0 hrs, 2.5 μM) and a test dose (time 12 hrs, 2.5 μM). Plotted are Fe$_{Lumen}$ (black), LIP (orange), FT core iron (red), Fe$_{Blood}$ (green), and Fe$_{body}$ (blue). (B) Two groups of nine time course simulations were run where a loading dose (time 0 hrs, 2.5 μM) and a test dose (time 12 hrs, 2.5 μM) were applied and the cumulative uptake and transferred iron recorded after 72 hrs. Each simulation had parameters identical to the control except for the value of the $k_{cat}$ rate constant of FT synthesis (IRPs regulated), which varied between 0.035 and 1.5 pMs$^{-1}$ (black lines) or the rate constant of LIP-induced DMT1 endocytosis LIP, which varied between 1 and 40 pM s$^{-1}$ (blue lines). Both parameters are represented in the abscissa as relative changes to the control value. The total iron uptake and transfer amounts are expressed as a percentage of the total iron dose (loading plus test dose, 5 μM, solid line for uptake, dashed line for transfer). (C) Time course of transferred iron amount (fmol) and core FT as a function of the FT synthesis rate constant. (D) Time course of transferred iron amount (fmol) and core FT iron as a function of the DMT1 endocytosis rate constant. (E) Time course of LIP amount as a function of the rate constant of FT synthesis. (F) Time course of LIP amount as a function of the rate constant of DMT1 endocytosis.

(-25%). Increasing the DMT1 endocytosis rate constant, on the other hand, results in only a minor decrease in iron uptake (-8%) and even smaller decrease in iron transfer (-3%).

## Discussion

Determining the quantitative impact of individual components of biological systems on observed phenomena is facilitated by the synergistic relationship between modeling and experimental studies. Computational modeling is useful in two major ways: it allows one to integrate data from independent experiments in a way that confirms (or not) the consistency of the mechanisms identified in the context of a larger system. This integrative aspect was used here in terms of putting together three mechanisms that were previously hypothesized as having a role in the mucosal block, and we used the model to quantify how important that role was in effect. Another major utility of computational models in supplementing laboratory experiments is their ability to isolate and modify a particular system component, while keeping the rest of the system unaltered. In laboratory experiments, particularly *in vivo*, it is hard or impossible to suppress various compensating responses of the system. In an experimental setting, when a particular cellular component is perturbed, compensatory mechanisms are exerted amidst a myriad of regulatory pathways to respond to the perturbation. With computational models it is possible to directly manipulate specific processes without allowing other compensatory effects, making it possible to dissect the specific effect of each process toward an observed behavior. We applied this approach to study the regulation of iron absorption in intestinal enterocytes using a computational model that was assembled based on knowledge of its components derived from extensive literature data.

The model presented here was constructed to quantify the relative quantitative importance of the mechanisms responsible for enterocyte iron mucosal block, a specific iron absorption behavior. However, we hope the model may additionally serve as a "model brick" [23] for future iron absorption models. Because this model was built for an explicit purpose, there are aspects of iron metabolic behavior still to be added that would be necessary to describe the totality of iron regulation by the duodenum. Thus, in addition to providing insights into mucosal block behavior, we hope the model here can be used as a starting point for future modeling efforts in the duodenum. Several potential future model additions and alterations are highlighted in the Assumptions section of the Methods to assist in doing so.

The present enterocyte mucosal block model was validated by showing that it reproduces experiments not used to calibrate it, and which were carried out in a variety of conditions including some very different to those that produced the data used to calibrate it. Additionally, the model reproduces well-known aspects of iron absorption, namely the importance of the size of the iron loading dose (ingestion of high amounts of iron [34]) and the hepcidin regulation of ferroportin-dependent iron export to the plasma. To this extent, the enterocytes are relatively passive elements, however they still contain mechanisms that modulate how much and how fast iron can be a) imported to their cytoplasm (uptake), and b) exported to plasma (transfer, which corresponds directly to iron absorption to the body). Because of this differential between uptake and transfer, and the relatively short lifetime of enterocytes, much iron can be taken up by enterocytes that is never absorbed (and is lost with enterocyte death and excretion). The two major intracellular mechanisms implicated are the amount of cytosolic ferritin (FT), which synthesis is regulated by the iron regulatory proteins (IRPs), and the endocytosis of the divalent metal transporter 1 (DMT1), which is induced by the labile iron pool (LIP). Following validation and the reproduction of known iron metabolic behavior, the model was used for simulations intended to dissect the role of mechanisms previously proposed to have an effect on the dynamics of enterocyte iron absorption: DMT1 endocytosis, FT iron sequestration, and the regulation of FT synthesis by IRPs.

Model simulations show that the largest impact on both iron uptake and transfer is mediated by the synthesis of FT, and that the DMT1 endocytosis has a minor effect on these two fluxes of iron. These results are consistent with what was previously proposed by Galy et al. 2013 [34], with addition of quantitative assignment of impact (FT vs DMT1) and further elucidation of cellular roles. The model here is able to disentangle the effect of each interrelated regulatory mechanism highlighting the specific aspects of IRPs and FT and their interactions that contribute to the iron transfer block. By simulating the model at different basal rates of FT expression and different strength of its control by IRPs, the model builds on the previous findings, suggesting that IRPs regulation is most impactful only in states where FT expression is relatively high. The model also shows how increased ferritin expression yields greater iron uptake capacity by enterocytes, but by increasing the cytosolic buffering capacity simultaneously decreases the iron transfer capability. Consequently, we show IRPs most significantly exert their effect when FT expression rate is high. These uptake and transfer effects are evidenced by the simulation results of Fig 4, the sensitivity analysis (Table 2), and its magnitude shown directly in comparison to that of DMT1 regulation (Fig 5B).

The high buffering capacity of FT also has consequences at the tissue-level given that duodenal enterocytes have short life span and are sloughed to the intestinal lumen at a considerable rate [104]; thus it is expected that such sloughing of enterocytes with high ferritin-bound iron content will result in irreversible loss of iron that had been uptake by enterocytes but will be irreversibly lost from being transferred to the rest of the body. To investigate such dynamics, however, is beyond the scope of this work and will require a multiscale model of the villus structure and dynamics.

Lastly, the model is able to shed light on the hypothesis that rapid regulation of apical membrane DMT1 concentration in enterocytes reduces the amount of iron that enterocytes can absorb [18]. Our model shows that varying the sensitivity of such endocytosis does indeed lead to lower total iron uptake (Fig 4A). However, the model predicts that while reduced apical DMT1 resulting from endocytosis reduces the total *uptake* over time, at the range of iron doses tested here, endocytosis does not appreciably reduce total iron *transfer* (Fig 4D). While this may lead one to question the significance of this phenomenon, our simulations indicate another important role DMT1 endocytosis may have. Faster rates of endocytosis serve to limit the transient peak of intracellular LIP (cellular free iron concentration) after an iron dose (Fig 5F). This effect is additional to the previously identified role of FT in buffering LIP levels [105]. While the exact mechanisms behind the regulation of DMT1 endocytosis are still being clarified, experiments have suggested that one driver connecting LIP levels and DMT1 endocytosis may be reactive oxygen species (ROS) [27]. Because LIP can catalyze ROS formation, it is possible that DMT1 endocytosis serves to protect the cells from ROS spikes, based on the LIP peak reduction prediction observed in these simulations.

## Methods

### Software

All simulations were carried out with COPASI versions 4.35 and 4.36 [24–26] on a Windows 10 computer with an Intel Core i7-4770 CPU at 3.40GHz. Some COPASI simulations were driven by Python scripts using the library BasiCO, a python API for COPASI [98]. Tasks were either run directly using the COPASI graphical user interface and collecting their output into .tsv files, or the task was run using the BasiCO scripts. Plots were created using Python's Matplotlib library, and exported to vector graphics SVG files and later refined using Inkscape (https://inkscape.org/). Experimental data were gathered from the supplemental materials

of the relevant publications, or digitized from their figures using an image digitizer (https://automeris.io/WebPlotDigitizer/). The process diagram to visualize the model (Fig 1) was created with Cell Designer [99] and converted to the SBGN format using Inkscape.

## Model Construction

**DMT1 iron transport and regulation.** To absorb inorganic dietary iron, duodenal enterocytes express divalent metal transporter 1 (DMT1) and duodenal cytochrome *b* (DCYTB) to their apical membrane [37]. DMT1 transports several divalent metals, including ferrous iron ($Fe^{2+}$). Before ferric iron ($Fe^{3+}$) from the diet can be transported through DMT1 it must be reduced to ferrous by DCYTB. DMT1 activity is regulated through endocytosis in a LIP-sensitive manner. The precise mechanistic details are not entirely known, but phenomenologically it has been shown that an increase in enterocyte LIP triggers endocytosis of DMT1 on the apical membrane (potentially driven by presence of reactive oxygen species [27]). Endocytosis reduces the density of (active) DMT1 proteins on the basolateral membrane and increases the cytosolic, vesicular DMT1 pool [18,19,27]. Ultimately, endocytosis may provide a negative feedback mechanism for transiently reducing the iron import flux through DMT1 when the cytoplasmic iron (LIP) levels are high. This mechanism does not reduce the total DMT1 protein level, only its activity by internalization. Because of the presence and importance of DMT1 in enterocyte iron absorption, the model presented here includes both DMT1 iron transport and endocytosis. DMT1 is represented by two species (for membranous and cytosolic DMT1) and six corresponding reactions. While the role of DCYTB in converting dietary ferric iron to ferrous is well known, we excluded DCYTB from the model because a block still occurs when enterocytes are exposed to ferrous iron. Thus while DCYTB may participate in a block when ferric iron is present, a simplified model excluding DCYTB is still sufficient to capture the block phenomena. The reactions are outlined in Table 3.

**Ferritin iron sequestration.** Ferritin (FT) is a cytosolic protein that forms a hollow protein cage made up of 24 subunits capable of sequestering 4300 iron atoms per cage [39–41]. The ability of FT to sequester such quantities of ferrous iron is necessary in maintaining appropriately low levels of LIP [39,41,42]. A FT cage stores excess LIP (soluble ferrous iron, $Fe^{2+}$) by importing and converting it into an insoluble ferric iron ($Fe^{3+}$) mineral core [43–45]. The storage process begins as a ferrous ion moves through pores in the FT cage into the cavity

**Table 3. DMT1 iron transport and regulation.** Shown are the reaction names, reactions, rate, laws, parameters and references for each process involved in DMT1 regulation and iron transport. The LIP independent DMT endocytosis represents endocytosis of DMT1 from the apical membrane and into the cytoplasm that occurs that is not driven by LIP. DMT1 fusion represents the reverse reaction; cytoplasmic DMT1 returning to the membrane. Both reactions are represented by mass action kinetics and are responsible in part for establishing the steady state apical cytoplasmic DMT1 fraction ratio [29]. The DMT1 iron transport reaction represents DMT1 facilitating ferrous iron movement from the lumen into the cytoplasm (increasing LIP). It is irreversible and represented by HMM kinetics with $Fe_{lumen}$ as the substrate. LIP dependent DMT endocytosis is represented phenomenologically by Hill-like kinetics. The last term in the reaction rate can vary between 0-1 and modifies the endocytosis rate based on LIP concentration.

| Reaction | Reaction | Rate Law | Parameter | Value | Ref. |
|---|---|---|---|---|---|
| **DMT1 endocytosis free** | $DMT1_{active} \rightarrow DMT1_{cytoplasm}$ | $k_{cat} \times DMT1_{active}$ | $k_{cat}$ | 29.4 s⁻¹ | – |
| **DMT1 endocytosis LIP modified** | $DMT1_{active} \rightarrow DMT1_{cytoplasm}$ | $k_{cat} \times DMT1_{active} \times \dfrac{LIP^n}{K_m^n + LIP^n}$ | $k_{cat}$ | 14.5 s⁻¹ | – |
| | | | $K_m$ | 2.805 M | – |
| | | | $n$ | 1.03 | – |
| **DMT1 fusion** | $DMT1_{cytoplasm} \rightarrow DMT1_{active}$ | $k_{cat} \times DMT1_{cytoplasm}$ | $k_{cat}$ | 50 s⁻¹ | – |
| **DMT1 iron transport** | $Fe_{upper} \rightarrow LIP$ | $k_{cat} \times DMT1_{active} \times \dfrac{Fe_{lumen}}{K_m + Fe_{lumen}}$ | $k_{cat}$ | 6844 s⁻¹ | [38] |
| | | | $K_m$ | 2.835 M | [38] |

within [46]. After entry, the ferrous iron participates in a series of reactions catalyzed by the FT subunits resulting in its oxidation to the ferric state [47,48] effectively trapping it within the cage. The ferric iron requires reduction before it can be released [49]. While mechanisms of iron release from FT are still being uncovered, the mineral iron within FT has been shown to be able to be liberated directly to the LIP by autophagic degradation of FT cages, regulated by nuclear receptor co-activator 4 (NCOA4) [50]. Within the ODE model presented here, FT is represented as a concentration (not individual cages with unique iron contents) that can be increased via expression and decreased by degradation. The iron stored within the total FT is represented as collective ferrous and ferric iron stores, a simplification that extends a previously published BioModels Database [51] model (MODEL2211030001) [22]. The process of iron sequestration is simplified into the first four reactions contained within Table 4. The last two reactions of Table 4 capture the process of mineral iron release through degradation of FT. Note that the rate law for core release via FT can be simplified mathematically to $k_{deg}$ times core. The rate law is presented in the unsimplified form because the rate law reflects the fact that the amount of core released when some portion of the FT population degrades is proportional to the population of cages "FT" and the amount of core within those cages ("core/ FT"). As an example of intended model behavior, if there was a time point where there was an average of 100 iron atoms of core per cage within a cell, it is expected that on average the degradation of 1 cage liberates 100 atoms. This behavior is what the rate law captures, albeit once simplified captures more opaquely.

**FPN iron export and regulation by plasma iron levels.** The iron export protein ferroportin (FPN) is a transmembrane protein that is the only known exporter of ferrous, cytosolic iron (LIP) [54]. In enterocytes, FPN is the conduit through which dietary iron that has entered the cell is transferred to the blood and ultimately the rest of the body. FPN

**Table 4. FT mathematical representation.** Shown below are the four FT iron sequestration and two degradation reactions in the model, their stoichiometries, rate laws, and rate laws parameter values. Nucleation and mineralization were parameterized using parameter estimation. The others (oxidation, reduction, and degradation reactions) are parameterized from published data.

| Reaction | Reaction | Rate Law | Parameter | Value | Ref. |
|---|---|---|---|---|---|
| Oxidation | $2\ LIP \rightarrow DFP$ | $\dfrac{k_{cat} \times \dfrac{H+rO}{24+rO} \times FT \times LIP^n}{Km^n + LIP^n}$ | $k_{cat}$ | $591\ \text{s}^{-1}$ | [44] |
|  |  |  | $K_m$ | $0.35\ \text{mM}$ | [52] |
|  |  |  | $n$ | $1.3$ | [52] |
|  |  |  | $rO$ | $2$ | [22] |
|  |  |  | $H$ | $24$ | [22] |
| Reduction | $DFP \rightarrow 2LIP$ | $k_{deg} \times DFP$ | $k_{deg}$ | $0.2605\ \text{s}^{-1}$ | [44,52] |
| Nucleation | $2DFP \rightarrow 4core$ | $k_{cat} \times DFP^2 \times FT \times \dfrac{L+rN}{24+rN}$ $\times \dfrac{Ki^n}{Ki^n + core^n}$ | $k_{cat}$ | $5\text{x}10^7\ \text{s}^{-1}$ | [22] |
|  |  |  | $K_i$ | $0.4615\ \text{mM}$ |  |
|  |  |  | $n$ | $4$ |  |
|  |  |  | $rN$ | $50$ |  |
|  |  |  | $L$ | $0$ |  |
| Mineralization | $DFP \rightarrow 2core$ | $\dfrac{k_{cat} \times DFP \times core}{Km + DFP} \times \dfrac{Ki^n}{Ki^n + core^n}$ $\times \dfrac{4300^m - apc^m}{4300^m}$ | $k_{cat}$ | $0.101564\ \text{s}^{-1}$ | [22] |
|  |  |  | $K_m$ | $5\text{x}10^{-6}\ \text{M}$ |  |
|  |  |  | $K_i$ | $4.6458\ \text{mM}$ |  |
|  |  |  | $n$ | $4$ |  |
|  |  |  | $m$ | $8$ |  |
| FT core release | $core \rightarrow LIP$ | $k_{deg} \times \left(\dfrac{core}{FT}\right) \times FT$ | $k_{deg}$ | $5.46\times10^{-6}\ \text{s}^{-1}$ | [53] |
| FT degradation | $FT \rightarrow$ | $k_{deg} \times FT$ | $k_{deg}$ | $5.46\times10^{-6}\ \text{s}^{-1}$ | [53] |

contains two iron binding sites where LIP iron can bind and be shuttled into the plasma [55]. The ferrous iron exported out of a cell through FPN is oxidized by membrane-bound ferroxidase hephaestin [56]. FPN and its activity are regulated transcriptionally, post transcriptionally, and its activity on the surface of the cell is controlled by plasma hepcidin, which in turn depends on the systemic level of iron and liver production. The binding of circulating hepcidin to FPN not only blocks the channel preventing iron export, but also triggers FPN internalization, ubiquitination, and subsequent degradation, reducing the active FPN at the membrane [57,58].

In the model presented here, and considering the purpose of analyzing aspects of cellular mucosal block, FPN activity serves as a "control" to establish biological plausibility and credibility. The role of hepcidin in controlling enterocyte iron export via FPN is well established. Here, hepcidin regulation of FPN is included so that the influence of the other mechanisms encoded by the model can be quantitatively compared to hepcidin's role in block of iron transfer to the body. Because enterocytes have been shown to have low basal levels of FPN expression, FPN IRE lacking mRNA, and the relative magnitude of the hepcidin control, in the model FPN is controlled by plasma iron levels alone and not IRPs. Because the inclusion of other organs is currently outside the scope of the model purpose and because is ultimately the plasma iron that drives hepcidin production in the liver, the entire systemic regulation process is simplified to one reaction, FPN inactivation, shown in detail with the other FPN related reactions in Table 5. The modeling decisions regarding the process for including systemic feedback are not trivial. Practically, FPN as a species is represented as a relative value scaled so that the iron transport matches experimental rates [59,60].

**IRPs regulation of FT, DMT1.** While differentiating and to some extent while mature, enterocytes utilize iron regulatory proteins 1 and 2 (IRP1 and IRP2, collectively referred to as IRPs) to post transcriptionally regulate several essential iron related proteins, including FT, FPN, and DMT1 [16]. IRPs bind sequences in the untranslated regions of mRNA called iron-responsive elements (IREs) [61–63]. The location and number of the IRE in the mRNA transcript determines the effect of IRPs binding. IRPs binding can either increase (3' IRE) or repress (5' IRE) translation. In turn, IRPs binding is inhibited by LIP. Thus in low iron conditions IRPs block and decrease synthesis of FPN and FT, which have 5' IREs [64–66], while stabilizing some DMT1 splice variants, which have 3' IREs, increasing their expression, especially in enterocytes [67]. Conversely, in iron excess, IRPs activity is inhibited by LIP, leading to higher levels of FPN and FT [17] and lower DMT1. IRP1 and IRP2, though similar in their binding activity, are regulated differently by LIP. IRP1 binds iron and undergoes a conformational change into a form incapable of efficiently binding IREs [68–70], while IRP2 is degraded in an iron dependent mechanism [71–73]. However, the model here represents IRP1/2 as one species "IRPs" that can be active or inactive. The reasoning is outlined in the

**Table 5. FPN iron export and regulation by plasma iron levels representation. Shown below are the FPN related reactions in the model, their stoichiometries, rate laws, and parameter values.**

| Reaction | Reaction | Rate Law | Parameter | Value | Ref. |
|---|---|---|---|---|---|
| **FPN-inactivation** | $FPN_{active} \rightarrow FPN_{inactive}$ | $k_{cat} \times FPN_{active} \times \dfrac{Fe_{blood}{}^{n}}{K_m{}^{n} + Fe_{blood}{}^{n}}$ | $k_{cat}$ | $1.44 \times 10^{-6}\ s^{-1}$ | – |
| | | | $K_m$ | 12.21 μM | – |
| | | | $n$ | 2.72 | – |
| **FPN-activation** | $FPN_{inactive} \rightarrow FPN_{active}$ | $k_{cat} \times FPN_{inactive}$ | $k_{cat}$ | $0.437 \times 10^{-12}\ cm^2\ s^{-1}$ | – |
| **FPN iron transport** | $LIP \rightarrow Fe_{blood}$ | $k_{cat} \times FPN_{active} \times \dfrac{LIP}{K_m + LIP}$ | $k_{cat}$ | $1.88\ s^{-1}$ | [59,60] |
| | | | $K_m$ | 2.31 μM | [55,59] |

Assumptions section below. Practically, the total IRPs concentration and resultant time course of FT and DMT1 are used to represent the IRPs effect in the model, so each isn't individually represented. A useful addition to the model outside the current scope would be to represent their individual concentrations and regulation explicitly. To incorporate the effect of IRPs, LIP modifies the total active IRPs through an "IRPs inactivation" reaction, and the reaction for FT expression is modified by active IRPs levels, both outlined in Table 6.

**Additional reactions.** In addition to the reactions represented in Tables 3–6, the model contains three additional reactions (Table 7). These reactions include rest of body sequestration of plasma iron, basal enterocyte iron uptake from plasma, and paracellular iron movement direct from lumen to plasma, reversibly. These reactions are all represented by mass action kinetics. Body sequestration was calibrated using the whole body model of Parmar *et al.* [75,76]. To do so, a range of initial plasma iron concentrations was fit to a linear model (of non-duodenum iron sequestration) and the slope variable of the model used as the mass action rate constant. The basal uptake and paracellular routes for iron uptake are also both simplifications of more complicated processes. Basal uptake in enterocytes, like many other mammalian cell types relies on transferrin (Tf) and transferrin receptor 1 (TfR1), and the endocytosis of the complex after plasma circulating Tf binds TfR1 on the basal membrane. While they may represent potentially more impactful processes in iron metabolism in other cell types, their contribution to total iron movement in the enterocyte model is minimal compared to other reaction fluxes [76]. The values of their respective rate constants were determined during parameter estimation, reflective of and validated by what has been seen experimentally.

**Model assumptions and species inclusion justification.** The present model is built for the explicit purpose of analyzing the dynamics of 1) DMT1 endocytosis, 2) FT iron buffering, and 3) IRPs regulation dependent FT expression on the iron handling behavior of a single mature enterocyte, specifically cellular mucosal block. Every modeling decision was motivated by that underlying goal, especially species and reaction selection/determination, to ensure a model with both minimal and sufficient detail. Because of this, the model is suited to address mucosal block dynamics, while there are several areas that should ideally be expanded in

**Table 6. IRPs mathematical representation. Shown below are the IRPs related reactions in the model, their stoichiometries, rate laws, and rate laws parameter values.**

| Reaction | Reaction | Rate Law | Parameter | Value | Ref. |
|---|---|---|---|---|---|
| **IRPs inactivation** | $IRPs_{active} \rightarrow IRPs_{inactive}$ | $k_{cat} \times IRPs_{active} \times LIP$ | $k_{cat}$ | 4 M$^{-1}$s$^{-1}$ | – |
| **IRPs activation** | $IRPs_{inactive} \rightarrow IRPs_{active}$ | $k_{cat} \times IRPs_{inactive}$ | $k_{cat}$ | $4.63 \times 10^{-6}$ s$^{-1}$ | – |
| **FT expression** | $\rightarrow FT$ | $k_{cat} \times \left(1 - \dfrac{IRPs_{active}^{n}}{K_m^n + IRPs_{active}^{n}}\right)$ | $k_{cat}$ | 0.0768 pMs$^{-1}$ | [74] |
|  |  |  | $K_m$ | 14 pM | [74] |

**Table 7. Additional reactions. Shown below are the additional three reactions in the model, their stoichiometries, rate laws, and rate laws parameter values. The reactions represent rest of body sequestration of plasma (blood) iron ("Body Sequestration"), basal membrane enterocyte iron uptake from plasma ("Basal Uptake"), and paracellular iron movement direct from lumen to plasma, reversibly ("Paracellular Iron Movement").**

| Reaction | Reaction | Rate Law | Parameter | Value | Ref. |
|---|---|---|---|---|---|
| **Body Sequestration** | $Fe_{Blood} \rightarrow Fe_{body}$ | $k_{cat} \times Fe_{Blood}$ | $k_{cat}$ | 0.000329 s$^{-1}$ | – |
| **Basal Uptake** | $Fe_{Blood} \rightarrow LIP$ | $k_{cat} \times Fe_{Blood}$ | $k_{cat}$ | 2.22 x 10$^{-16}$ s$^{-1}$ | – |
| **Paracellular Iron Movement** | $Fe_{upper} \leftrightarrow Fe_{Blood}$ | $k_{for} \times Fe_{upper} - k_{rev} \times Fe_{Blood}$ | $k_{for}$ | 3.88 x 10$^{-22}$ s$^{-1}$ | – |
|  |  |  | $k_{rev}$ | 3.17 x 10$^{-15}$ s$^{-1}$ | – |

future model iterations of more general iron metabolism. The specific assumptions of the model not explained in the previous sections are described below, including justifications of why iron species essential to enterocyte iron metabolism are not included. The assumptions also serve to highlight potential future modeling additions/extensions for future model applications, building toward a "complete" *in silico* model of duodenal enterocyte iron regulation.

1. **Protein synthesis.** For all protein species, transcription and translation are grouped into one reaction "protein synthesis" that increases the protein concentration in one step. Motivation for the one reaction grouping is two-fold. First, the grouping still captures enough level of detail that the process can be altered where biologically indicated while simplifying the model and reducing the number of parameters. Second, there is evidence that in mature enterocytes at the villus tip, translation occurs at a relatively higher rate than transcription [19,38,77]. The importance of translational regulation versus transcriptional regulation is reinforced by the reliance of enterocytes on IRPs to regulate protein levels post-transcriptionally. Additionally, the grouping allows for the modification of protein synthesis by IRPs.

2. **Heme iron.** Apical uptake and subsequent metabolism of heme is excluded from the model. While heme is a necessary dietary component for many mammals and thus important to include in a model attempting to capture iron regulation in full, it is excluded from the model for a few reasons. Essential players in enterocyte uptake of heme are still being identified, so there is not enough detail to include a complete pathway. Also, the mucosal block the model presented here characterizes occurs in enterocytes exposed to inorganic iron. Thus, while the presence of heme in the lumen undoubtedly modifies the iron metabolism of enterocytes and it would in future iterations ideally be included, the simplest form of mucosal block occurs even in its absence, meaning a model without heme still contains sufficient detail to capture the effect.

3. **DMT1 and DCYTB.** The DMT1 iron import reaction is a grouping effect of the activity of DMT1 and DCYTB. DCYTB is a reductase associated with DMT1 that converts ferric iron in the lumen into a ferrous form to allow DMT1 to transport it and is thus critical to iron absorption. However, DCYTB is excluded for several reasons. The main reason is the presence of mucosal block in ferrous iron exposed enterocytes, backed by a significant number of experiments using ferrous iron. Similar to exclusion of heme, not including DCYTB allows the disentanglement of the availability of iron dependent on meal composition from DMT1 import and endocytosis dynamics. In other words, instead of having to incorporate a set of lumen iron availability reactions that depends on DCTYB expression and reaction kinetics, the concentration of ferrous iron added to solution (or in a meal) can be used directly. Also adding credibility to its exclusion is the evidence that DCYTB knockout does not induce a notable iron phenotype, suggesting ferrireductase redundancy [78].

   In addition to the activity of DCYTB, there are several other factors that also modify the activity of DMT1. These modifiers include temperature, luminal pH, presence of other metals, etc. For simplicity and again due to the presence of a mucosal block under unvaried conditions of these variables, these additional modifiers are also not included. To test the effect of variations in DCYTB and these other modifiers by adding them to the model presented here would be useful future work, but is tangential to the model purpose here.

4. **FPN and HEPH.** There is a similar but not identical reasoning for the exclusion of the oxidase associated with FPN, hephaestin (HEPH) to the exclusion of DCYTB. While a mutation in hephaestin is known to result in accumulation of non-heme iron in enterocytes and in a

reduced transfer of iron to plasma [102], it appears that it is responsive to global levels of iron rather than the level of iron in the enterocyte [103], and as such we assume that it does not have a role in the mucosal block caused by consecutive dietary iron doses. The pathological iron mucosal block caused by the HEPH mutation is not addressed in this model.

5. **IRPs effect on DMT1/FPN translation.** The effects on protein synthesis IRP1 and IRP2 have on DMT1 and FPN are not included. In the model, DMT1 the endocytosis alone determines the active DMT1 concentration [19]. There are several reasons why the IRPs effect on DMT1 mechanism, though undoubtedly present in duodenal enterocytes, is not included.

In contrast to FT, for DMT1 the physiologic role of IRPs post-transcriptional regulation, while extensively described, is still being established. Also still being worked out is the relative importance of IRPs regulation in determining apical DMT1 levels compared to relocation (endocytosis) regulation in terminally differentiated enterocytes. Total cellular DMT1 protein concentration depends on 1) basal transcription and 2) basal translation rates, 3) transcriptional rate modification (including of the DMT-1A variant by HIF2-alpha [79,80] which in turn is also regulated by IRPs [34]), 4) post-transcriptional regulation by IRPs binding (complicated by the fact that DMT1 has splice variants expressed by enterocytes that do not contain an IRE [34,81]), and 5) mRNA and protein degradation rate. Layered on top of that whole cell concentration regulation is post-translational cellular location regulation. The DMT1 fraction on the apical membrane is ultimately what transports ferrous iron into the enterocyte, meaning that while transcriptional and translational regulation are necessary for modifying total DMT1, DMT1 cellular location regulation determines uptake. Experimentally, there are data describing the dynamics across all these DMT1 regulation levels. Absence of an DMT1 +IRE variant leads to iron deficiency in mice and it has been shown that a decrease in IRPs activity following iron exposure leads to lower DMT1 +IRE mRNA stability [20] and mRNA concentration (50%) [19]. While lower mRNA levels tend to be associated with lower protein levels, such a phenomenological simplification is not always seen. Galy et al. 2013 [34] showed an increase in DMT1 protein in IRP-deficient mice, despite the prediction of the opposite effect. There have also been multiple studies that attempt to describe not just the change in transcription and translation, but location of DMT1 protein as well (apical membrane versus whole cell). In these studies [4,14,19,27,28,82], when enterocytes are exposed to iron, the membrane DMT1 levels decrease, but the changes to total DMT1 (whole cell) are relatively minor (shown by a subset of studies that measure whole cell concentration simultaneously [4,19,82]). This suggests that while IRPs may modestly modify total DMT1 levels in terminally differentiated villus enterocytes, we make the assumption that excluding explicit IRPs impact will not affect the membrane component significantly relative to impact of DMT1 relocation.

Considering the multiple levels of DMT1 regulation data, the model excludes the effect of IRPs on DMT1 expression. However, a version of the model was created that includes IRPs regulation of DMT1 and that model output was compared to that of the one here. Ultimately, the results with or without the IRPs regulation show only minor differences in the five iron uptake characteristics we establish for analysis in the results section, but the general trends regarding parameter sensitivity in the model are not altered, indicating that while one version of the model may be more accurate than the other, the conclusions made congruent. Proper experimental work disproving this would be to remove IRE-sensitive expression of DMT1 and see if block is removed.

For FPN, the exclusion of IRPs directed effect is driven by the purpose of the model here, namely describing the reduction in iron exported out of FPN due to mucosal

block. As a result, the necessary FPN details are its iron transport kinetics and basal membrane fraction, with additional detail, though potentially biologically correct, irrelevant to the current scope and excluded for simplicity. Also, the model relies on the fact that even if IRPs changes the FPN total, hepcidin levels ultimately determine the total active FPN fraction. For future models, there should still be deliberation on relevance of IRPs directed FPN change before addition to the model, as it appears that there is greatly reduced IRPs driven FPN expression in duodenal enterocytes [15,83,84] compared to other cell types.

6. **IRP1 and IRP2** There are two distinct iron regulatory proteins (IRPs), IRP1 and IRP2 that both post-transcriptionally regulate several iron related proteins. IRP1 and IRP2 have similar IRE binding activity, but are in turn regulated differently by LIP. Rising LIP levels drive IRP1 conformational change reducing its ability to bind IREs [68–70], while they cause IRP2 to undergo proteasomal degradation [71–73]. It is possible that because of the two distinct LIP regulated mechanisms, varying the total IRPs composition of IRP1 and IRP2 may result in varied total IRPs activity temporal dynamics. However, the model here represents IRP1 and IRP2 as one species "IRPs" that can be active or inactive. The motivation for the single species simplification is in part that these proteins have similar interaction with IREs [71,85] and redundancy in their activity [86,87]. Thus the total concentration of both is assumed to correlate to the total post-transcriptional effect. Experimentally, when both IRP1 and IRP2 are knocked out in the mouse intestine, the result is fatal, indicating that inclusion of the IRPs mechanism is essential [20,86]. However, in support of our simplification, single IRP1 or IRP2 knockout mice are viable [86,88–91]. The single species assumption is not made lightly. There is an abundance of data surrounding the differential regulation of IRP1 versus IRP2 by LIP and their binding activity [34,63,65,68,92–97]. However, efforts we made to construct a submodel focused on the differences between these species utilizing these data, revealed there were key elements missing from the literature, ultimately requiring our use of the approximation implemented here. Relevant data to properly build and IRP1/2 submodel would describe dose dependence changes on the activity of IRP1 and degradation IRP2 and the delineate of protein synthesis and concentration changes into the underlying transcription and translation rates of the IRPs regulated proteins specifically within terminally differentiated enterocytes (ideally also shown to be iron dose dependent in their changes).

## Supporting information

**S1 Text. Full set of ordinary differential equations of the model.**
(PDF)

**S2 Text. Full set of parameters used in the model.**
(PDF)

**S3 Text. Analysis of blocking magnitude dependence on iron blocking dose.**
(PDF)

**S1 Data. Models in COPASI format and data used for parameter estimation.**
(ZIP)

## Acknowledgments

We thank Drs. M. Blinov, S. Torti, and P. Vera-Licona for discussions about this project. We thank the UConn School of Medicine for supporting JM.

## Author contributions

**Conceptualization:** Joseph Masison, Pedro Mendes.

**Formal analysis:** Joseph Masison, Pedro Mendes.

**Funding acquisition:** Pedro Mendes.

**Methodology:** Joseph Masison.

**Project administration:** Pedro Mendes.

**Software:** Joseph Masison, Pedro Mendes.

**Supervision:** Pedro Mendes.

**Validation:** Joseph Masison.

**Visualization:** Joseph Masison.

**Writing – original draft:** Joseph Masison, Pedro Mendes.

**Writing – review & editing:** Joseph Masison, Pedro Mendes.

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
