## [Decision Letter · Decision Letter 0]

23 Oct 2024

PCOMPBIOL-D-24-01283Mathematical modeling reveals ferritin as the strongest cellular driver of dietary iron transfer block in enterocytesPLOS Computational Biology Dear Dr. Mendes, Thank you for submitting your manuscript to PLOS Computational Biology. After careful consideration, we feel that it has merit but does not fully meet PLOS Computational Biology's publication criteria as it currently stands. Therefore, we invite you to submit a revised version of the manuscript that addresses the points raised during the review process. In particular, the reviewers raise several points related to various biological aspects: how they are implemented, their relevant kinetics, and the impact on enterocytes. Other points related to parameter values and model assumptions are also mentioned. Please submit your revised manuscript within 60 days Dec 23 2024 11:59PM. If you will need more time than this to complete your revisions, please reply to this message or contact the journal office at ploscompbiol@plos.org. Please include the following items when submitting your revised manuscript: * A rebuttal letter that responds to each point raised by the editor and reviewer(s). You should upload this letter as a separate file labeled 'Response to Reviewers'. This file does not need to include responses to formatting updates and technical items listed in the 'Journal Requirements' section below.* A marked-up copy of your manuscript that highlights changes made to the original version. You should upload this as a separate file labeled 'Revised Manuscript with Track Changes'.* An unmarked version of your revised paper without tracked changes. You should upload this as a separate file labeled 'Manuscript'. If you would like to make changes to your financial disclosure, competing interests statement, or data availability statement, please make these updates within the submission form at the time of resubmission. Guidelines for resubmitting your figure files are available below the reviewer comments at the end of this letter. We look forward to receiving your revised manuscript. Kind regards, Stacey D. Finley, Ph.D.Section EditorPLOS Computational Biology Feilim Mac GabhannEditor-in-ChiefPLOS Computational Biology Jason PapinEditor-in-ChiefPLOS Computational Biology  **Journal Requirements:** **Additional Editor Comments (if provided):****Reviewers' comments:** Reviewer's Responses to Questions

**Comments to the Authors:**

Reviewer #1: An interesting paper that uses mathematical modeling in an effort to define duodenal intestinal iron uptake and the mucosal block in enterocytes. The authors have an in depth understanding of intestinal iron metabolism and provide reasonable rationale for the simplification of the processes involved in iron transport. This reviewer is not competent to assess the modeling component so not able to assess the validity of the modeling approaches.

Most of the questions that arose during review of the manuscript were subsequently covered in the assumptions section of the Methods where the authors go into some detail about specific components that they decided to include and exclude from the model. The authors may want to consider a short paragraph – perhaps in the results – where they summarize the assumptions and refer the readers to the methods for further information. Some of the assumptions, such as Hephaestin not contributing to mucosal block, may not be entirely accurate as a key finding in mice mutant in this gene is enterocyte accumulation of iron in ferritin to very high levels – a constitutive mucosal block. However, as this represents a non-physiologic state, it is reasonable to exclude from the model as they do.

One possible contribution to the regulation of intestinal iron uptake that the authors do not explicitly discuss is the role of enterocyte sloughing and turnover which has been suggested by some authors to play a role in preventing the iron bound in ferritin as a result of the mucosal block in ever being absorbed.

It would be interesting to consider the kinetics of their model with the kinetics of intestinal cell turnover – especially in the 48-72 hour period after a mucosal blocking dose when new enterocytes will likely constitute a significant portion of the transporting enterocytes and similarly a portion of the iron bound to ferritin in the enterocytes will be lost by turnover. This reviewer wonders whether taking this turnover into account would improve the model. Is the iron in Ferritin in the enterocyte – a dead end – in that it is unlikely to be mobilized within the time scale of the enterocyte lifespan or alternatively could it serve to buffer iron uptake as they suggest for DMT1 to prevent excess iron uptake.

The authors also don't explicitly discuss the role of very high duodenal iron leading to NTBI in the enterohepatic circulation – clearly the mucosal block is not effective at some threshold and is there any data in the literature that could give some insight into this phenomenon, e.g. when does the model they provide breakdown - especially if a potential use of the model is to help develop/understand physiologic iron uptake or an overall model of iron homeostasis as a building block. It is important to understand the intestinal iron doses where the model is appropriate to utilize and when the model is likely to be violated.

Reviewer #2: see attachment

Reviewer #3: Review Paper PCOMPBIOL-D-24-01283

Mathematical modeling reveals ferritin as the strongest cellular driver of dietary iron transfer block in enterocytes

Summary

This paper presents a mathematical model that describes the processes of iron uptake and transfer in enterocytes. The model focuses on several key phenomena, including the endocytosis of the DMT1 transporter, the role of ferritin, and its post-transcriptional regulation of FT expression by iron regulatory proteins (IRPs), as well as the regulation of ferroportin by hepcidin. The model has been calibrated and validated using experimental data available in the literature.

The primary goal of this model is to clarify which mechanisms are mainly responsible for mucosal block. The authors conclude that the key factor is the role of ferritin in iron transfer. Additionally, the endocytosis of DMT1 plays an important role in iron uptake.

Main concerns

In https://doi.org/10.1186/1752-0509-4-147 it was presented a model that “predicts desirable characteristics for a buffer protein such as effective removal of excess iron, which keeps intracellular cLIP levels approximately constant even when large perturbations are introduced, and a freely available source of iron under iron starvation. In addition, the simulated dynamics of the iron removal process are extremely fast, with ferritin acting as a first defense against dangerous iron fluctuations and providing the time required by the cell to activate slower transcriptional regulation mechanisms and adapt to iron stress conditions.” Please correct some claims in the manuscript accordingly.

There is no discussion regarding the relative characteristic time of each phenomenon considered in the model and its impact on the construction of the model and subsequent conclusions. This is relevant considering the long simulation ranges up to 72 h and the different regulation systems that are in place: systemic, transcriptional, translational and mucosal block, with response times varying from days to minutes after an iron challenge. Hepcidin regulation is considered, but DMT1 and FPN regulation are not included in the model. This could have a strong impact on the results of the model, given the long simulation times.

The data from Esparza (2015) indicate that the kinetics of DMT1 endocytosis are rapid and vary depending on the type of iron administered. The simulations presented in Figures 2B and 2D do not adequately represent the complexity of the data from the original studies. Specifically, in Figure 2B, the model depicts an exponential decrease in the amount of DMT1 on the membrane that continues for up to 120 minutes of simulation. However, the data from Esparza (2015) show a reappearance of DMT1 on the apical membrane after 20 minutes, with the rate of reappearance depending on the type of iron.

The data from Colins (2017) indicate that the displacement of transporters or mucosal block can occur in less than 30 minutes. The experimental results reveal changes in the slopes of absorbed iron that the model does not account for. In fact, the model presents a linear approximation of iron trajectories, which may be an oversimplification. This is evident even in sequences as short as 8 to 10 minutes: “Between 3 and 12 min after the iron exposure, there is a significant decrease in the rate of iron absorption compared to its initial value. Nevertheless, during the next five minutes, the rate increases again. This behavior is observed for the three initial apical iron concentrations. The experimental patterns observed in the absorption rates over time for the three extracellular iron con-centrations studied drift away from the standard behavior of a transport system that could be described using a Michaelis–Menten or Hill type of expression. This behavior can be attributed o the variation in the amount of DMT1 present in the apical membrane after the iron expo-sure, as suggested by Nuñez et al”

Based on this statements such as “Without modifying any parameters, other than the initial conditions reflective of each experiment, the model adequately reproduced the observations of the validation experiments (Fig. 2D-F).” may not be fully accurate..

Line 212 Why is the steady state calculated in this manner? Is a stationary state of this kind reasonable from a biological perspective, given that the availability of iron in the apical compartment is rarely constant? Additionally, other regulatory systems operate on different time scales, and these are not accounted for in the model.

Line 219 Please explain the rationale for choosing a concentration of 125 nM in the intestinal lumen and why it should be so high to indicate iron overload.

Line 252 Including the experimental data in the figure would help evaluate the simulation results more effectively.Fig. 6 of Frazer et al. 2003 [15]

Line 329 The manuscript presents numerical simulations illustrating the effects of Kcat in LIP-induced DMT1 endocytosis and Kcat and Km in the synthesis of FT. However, it does not provide a mechanistic explanation based on the structure of the mathematical model or the behavior of the biological system. Please improve the critical discussion..

Line 297 There is a maximum in the Minimum Transfer Value plot Fig4N. Please discuss and explain why.

Line 412 Why a 12 h is chosen as the fixed-dose interval ? please explain in the manuscript.

Lines 457-495 The discussion section requires substantial enhancement. Specifically, the content situated between these lines resembles an extended summary rather than a critical analysis of the paper’s findings.

Line 479 Figure 2 illustrates that the model was calibrated using data within the range of 0 to 180 minutes. However, most of the simulations and conclusions are based on a range of 0 to 72 hours. Please discuss the impact of this discrepancy on the quality of the model and the relevance of the simulation results and conclusions.

Line 652: Additional reactions are mentioned; however, the equations and parameters are not provided. Please include all differential equations, initial conditions, and model parameters in a mathematical format that is not dependent on the COPASI platform in the supplementary material.

Minor comments

Line 374 possible typo: R instead of epsilon

Line 417 possible typo: 48 instead of 24? At 24 hours, the body’s iron is increasing, and the LIP has a substantial amount of iron.

Line 626 possible typo: space between FP_N

**Have the authors made all data and (if applicable) computational code underlying the findings in their manuscript fully available?**

Reviewer #1: Yes

Reviewer #2: Yes

Reviewer #3: Yes

PLOS authors have the option to publish the peer review history of their article (what does this mean? ). If published, this will include your full peer review and any attached files.

**Do you want your identity to be public for this peer review?** For information about this choice, including consent withdrawal, please see our Privacy Policy .

Reviewer #1: No

Reviewer #2: No

Reviewer #3: No

 **Figure resubmission:**While revising your submission, please upload your figure files to the Preflight Analysis and Conversion Engine (PACE) digital diagnostic tool, https://pacev2.apexcovantage.com/. PACE helps ensure that figures meet PLOS requirements. To use PACE, you must first register as a user. Registration is free. Then, login and navigate to the UPLOAD tab, where you will find detailed instructions on how to use the tool. If you encounter any issues or have any questions when using PACE, please email PLOS at figures@plos.org. Please note that Supporting Information files do not need this step. If there are other versions of figure files still present in your submission file inventory at resubmission, please replace them with the PACE-processed versions. 
---

## [Decision Letter · Decision Letter 1]

3 Feb 2025

Dear Prof. Mendes,

We are pleased to inform you that your manuscript 'Mathematical modeling reveals ferritin as the strongest cellular driver of dietary iron transfer block in enterocytes' has been provisionally accepted for publication in PLOS Computational Biology.

Best regards,

Stacey D. Finley, Ph.D.

Section Editor

PLOS Computational Biology

Alison Marsden

Academic Editor

PLOS Computational Biology

Reviewer's Responses to Questions

**Comments to the Authors:**

Reviewer #2: The authors have responded appropriately to my concerns. I recommend that the paper be accepted for publication.

Reviewer #3: Thank you for all the corrections.

**Have the authors made all data and (if applicable) computational code underlying the findings in their manuscript fully available?**

Reviewer #2: Yes

Reviewer #3: Yes

PLOS authors have the option to publish the peer review history of their article (what does this mean? ). If published, this will include your full peer review and any attached files.

**Do you want your identity to be public for this peer review?** For information about this choice, including consent withdrawal, please see our Privacy Policy .

Reviewer #2: No

Reviewer #3: No

---

## [Editor Report · Acceptance letter]

PCOMPBIOL-D-24-01283R1

Mathematical modeling reveals ferritin as the strongest cellular driver of dietary iron transfer block in enterocytes

Dear Dr Mendes,

I am pleased to inform you that your manuscript has been formally accepted for publication in PLOS Computational Biology. Your manuscript is now with our production department and you will be notified of the publication date in due course.

With kind regards,

Anita Estes
